# Estimating excess mortality during the COVID-19 pandemic from a population-based infectious disease surveillance in two diverse populations in Kenya, March 2020-December 2021

Clifford Oduor[1]*, Allan Audi[1], Samwel Kiplangat[1], Joshua Auko[1], Alice Ouma[1], George Aol[1], Carolyne Nasimiyu[2], George O. Agogo[3], Terrence Lo[3], Peninah Munyua[3], Amy Herman-Roloff[3], Godfrey Bigogo[1], Patrick K. Munywoki[3]

1 Center for Global Health Research, Kenya Medical Research Institute (KEMRI), Kisumu, Kenya, 2 Washington State University (WSU) Global Health Kenya, Nairobi, Kenya, 3 Division of Global Health Protection, Global Health Center, Centers for Disease Control and Prevention, Nairobi, Kenya

* cooduor@kemri.go.ke

## Abstract

Robust data on the impact of the COVID-19 pandemic on mortality in Africa are relatively scarce. Using data from two well-characterized populations in Kenya we aimed to estimate excess mortality during the COVID-19 pandemic period. The mortality data arise from an ongoing population-based infectious disease surveillance (PBIDS) platform, which has been operational since 2006 in rural western Kenya (Asembo, Siaya County) and an urban informal settlement (Kibera, Nairobi County), Kenya. PBIDS participants were regularly visited at home (2–3 times a year) by field workers who collected demographic data, including deaths. In addition, verbal autopsy (VA) interviews for all identified deaths are conducted. We estimated all-cause and cause-specific mortality rates before and during the height of the COVID-19 pandemic, and we compared associated mortality rates between the periods using incidence rate ratios. Excess deaths during the COVID-19 period were also estimated by modelling expected deaths in the absence of COVID-19 by applying a negative binomial regression model on historical mortality data from January 2016. Overall and monthly excess deaths were determined using the P-score metric. Spearman correlation was used to assess whether there is a relationship between the generated P-score and COVID-19 positivity rate. The all-cause mortality rate was higher during the COVID-19 period compared to the pre-COVID-19 period in Asembo [9.1 (95% CI, 8.2–10.0) vs. 7.8 (95% CI, 7.3–8.3) per 1000 person-years of observation, pyo]. In Kibera, the all-cause mortality rate was slightly lower during the COVID-19 period compared to the pre-COVID-19 period [2.6 (95% CI, 2.2–3.2 per 1000 pyo) vs. 3.1; 95% CI, 2.7–3.4 per 1000 pyo)]. An increase in all-cause mortality was observed (incidence rate ratio, IRR, 1.16; 95% CI, 1.04–1.31) in Asembo, unlike in Kibera (IRR, 0.88; 95% CI, 0.71–1.09). The notable increase in mortality rate in Asembo was observed among persons aged 50 to 64 years (IRR, 2.62; 95% CI, 1.95–3.52), persons aged 65 years and above (5.47; 95% CI, 4.60–6.50) and among females (IRR,

**Data Availability Statement:** The data can be found in the Harvard Dataverse: https://doi.org/10.7910/DVN/ASOVJJ.

**Funding:** Funding for this study was provided by the United States Centers for Disease Control and Prevention, through a Cooperative agreement # 6 NU2HGH000031 with Washington State University and Kenya Medical Research Institute awarded to GB. The specific roles of the authors are articulated in the 'author contributions' section. The funders had no role in study design, data collection and analysis, decision to publish, or preparation of the manuscript.

**Competing interests:** The authors have declared that no competing interests exist.

1.25; 95% CI, 1.07–1.46). These age and gender differences were not observed in Kibera. We observed an increase in the mortality rate due to acute respiratory infection, including pneumonia (IRR, 1.45;95% CI, 1.03–2.04), and a reduction in the mortality rate due to pulmonary tuberculosis (IRR, 0.22; 95% CI, 0.05–0.87) among older children and adults in Asembo. There was no statistically significant change in mortality rates due to leading specific causes of death in Kibera. Overall, during the COVID-19 period observed deaths were higher than expected deaths in Asembo (P-score = 6.0%) and lower than expected in Kibera (P-score = -22.3%).Using well-characterized populations in the two diverse geographic locations, we demonstrate a heterogenous impact of the COVID-19 pandemic on all-cause and cause-specific mortality rates in Kenya. We observed more deaths than expected during the COVID-19 period in our rural site in western Kenya contrary to the urban site in Nairobi, the capital city in Kenya.

## Introduction

Mortality surveillance provides essential information for formulating evidence-based monitoring and response during pandemics such as coronavirus disease 2019 (COVID-19) [1]. Uncoded COVID-19 and other pandemic-related deaths can be captured by estimating excess mortality, thereby highlighting the full burden of the pandemic [2]. Robust surveillance systems capable of detecting variations in mortality at the community level can provide an unbiased and independent insight into the impact of epidemics within a country [3, 4]. These surveillance systems may be critical when high numbers of deaths occur outside of health facilities and particularly when health systems become overwhelmed [5]. In low and middle-income countries (LMICs) where the vital registration systems are suboptimal, population-based surveillance is an important tool for mortality surveillance and epidemic monitoring [6]. Resource-poor-settings often lack accurate, complete and timely reporting of in-hospital deaths therefore Verbal autopsy (VA) enables identification of cause of death in such situations where robust routine systems are lacking and where many people die at home [7].

Since the emergence of COVID-19, there has been evidence of substantially more morbidity and mortality globally [8]. In sub-Saharan Africa and more specifically in Kenya, the impact of the pandemic on mortality has been largely unquantified [9]. The first COVID-19 case in Kenya was confirmed on 12th March 2020 [10] and as of mid-February 2022 approximately 322,517 cases had been confirmed and 5,632 deaths reported [11]. Prevention measures ranging from physical distancing, movement restrictions, closure of schools, sanitation measures, testing, and wearing of face masks in public places have been introduced to avert COVID-19 morbidity and mortality. Higher COVID-19 vaccination coverage levels have been associated with reduced mortality rates [12]. In Kenya, COVID-19 vaccination commenced in early March 2021, with the first phase focused on vaccinating frontline health workers and by mid-February 2022, the vaccination coverage among persons aged 15 years and above was 26.4% [13]. By the end of 2021, there had been five discrete COVID-19 waves,: May 2020–August 2020, October 2020–January 2021, March 2021-April 2021, July 2021-September 2021, and the fifth wave from mid-December 2021-January 2022 [14]. In Kenya, the overall crude death rate was reported to be 5.3 per 1,000 people in the year 2020, down from 5.4 per 1,000 people the previous year [15]. To determine the impact of COVID-19 pandemic in Kenya, we used data from two diverse, well-characterized populations in a rural setting and an urban informal

settlement in Kenya to compare mortality patterns pre-and during the COVID-19 pandemic period and estimated the excess mortality that could either be directly or indirectly related to the COVID-19 pandemic.

## Materials and methods

### Study site and population

The Kenya Medical Research Institute (KEMRI) in collaboration with the U.S. Centers for Disease Control and Prevention (CDC) have conducted a longitudinal Population-Based Infectious Disease Surveillance (PBIDS) since 2006 in two geographically diverse sites, rural and urban, in Kenya [16, 17]. The rural site is in Asembo within Siaya county in western Kenya while the urban site is located in Kibera, the largest informal settlement in Nairobi County [18]. As of mid-2021, Asembo had ~35,000 PBIDS participants while Kibera had ~23,000 PBIDS participants under follow up. Being a rural site, Asembo consists of more elderly persons compared to Kibera which has relatively high number of younger people who have migrated to the city in search of work. Regular (2–3 times a year) household visits were performed to collect demographic and health data including residence status and deaths. Verbal autopsy (VA) interviews for all deaths are conducted within a month from the time of death by trained field workers. The deceased's next of kin, the health worker who cared for the person, or relatives who are familiar with the circumstances of the death are actively followed at home for the interview using WHO-approved VA questionnaires [19]. Several visits are made to the households in order to ensure that credible respondents are interviewed.

### Statistical analysis

PBIDS data, including all registered deaths from participants of all age groups between 1st January 2016 and 31st December 2021, were analyzed. Analysis of the data was performed using R for Windows version 4.1.2 [20] and STATA Version 17 software (Stata Corp., College Station, TX, USA). Assignment of probable cause(s) of death (CoD) was done using a Bayesian probabilistic model, the InterVA version 5 [21]. The CoD generated by InterVA-5 are compatible with the International Classification of Diseases version 10 (ICD-10) and utilizes outputs from version 1.5.3 of the electronic (Open Data Kit) 2016 WHO VA instrument [22]. We generated all-cause mortality rates which were expressed as the number of deaths of participants per 1,000 person-years of observation (pyo) calculated based on each PBIDS participant's residency status during the study period. The cause-specific mortality rates accounted for the proportion of deaths with missing VA data. Cause-specific fractions were determined as the proportion of all deaths that were attributable to a specific CoD. We categorized the study period into two time periods, pre-COVID-19 and during the COVID-19 pandemic period, delimited by 1st March 2020 (the month when the first COVID-19 case was confirmed). We compared the overall, age and gender-specific all-cause mortality rates between the periods using incidence rate ratios. We also compared cause-specific fractions pre-COVID-19 and during the COVID-19 period. To estimate excess deaths due to all causes, we modelled expected deaths in the absence of COVID-19 (counterfactual scenario) for the 'COVID-19 period' by applying a negative binomial regression model while adjusting for seasonality, after assessing for overdispersion, on historical mortality data collected between 1st January 2016 to 29th February 2020.

Overall and monthly excess deaths was assessed by generating the P-score metric which was calculated by getting the absolute number of deaths above or below expected deaths in the absence of COVID-19 (derived from the historic number of overall and monthly deaths)

divided by expected deaths multiplied by 100 [8]. A P-score of 100% would mean the observed death counts doubled the projected death counts. P-score is defined as:

P-score = [(observed deaths during COVID-19 period-expected deaths in absence of COVID-19)/ expected deaths in absence of COVID-19] *100

In addition, using data between 1st May 2020 to 31st December 2021 from individuals tested for COVID-19 at the outpatient clinics centrally located in each surveillance sites, we generated monthly SARS-CoV-2 PCR-positivity expressed as a percentage of the positive cases among the individuals tested. Details on the SARS-CoV-2 testing methods are described elsewhere [23]. Spearman correlation was used to assess whether there is a relationship between the generated p-score and COVID-19 positivity rate.

## Ethical considerations

Household heads provided written informed consent for their members to participate in PBIDS. The PBIDS protocol and consent procedures, including surveillance and VA activities, were approved by KEMRI scientific and ethics committee (protocol SSC#2761,) and U.S. CDC (#6775) Institutional Review Boards.

## Results

### Mortality data

In the pre-COVID-19 period (1st January 2016 to 29th February 2020), there were 285 deaths in Kibera and 1,025 deaths in Asembo. During the COVID-19 period (1st March 2020 to 31st December 2021) there were 118 deaths in Kibera and 445 deaths in Asembo. On average, 246 and 69 deaths were recorded per year in Asembo and Kibera, respectively, in the pre-COVID-19 period. The corresponding average deaths during the COVID-19 period were 243 and 64 per year in Asembo and Kibera (Table 1). Of the deaths in the pre-COVID-19 period, 160 (56.2%) were male and 78 (27.4%) were children under 5 years in Kibera. During the COVID-19 period, 70 (59.3%) were male and 28 (23.7%) were children under 5 years. In Asembo, of the deaths in the pre-COVID-19 period 516 (50.3%) were female and 190 (18.5%) were children under 5 years. During the COVID-19 period 229 (51.5%) were female and 72 (16.2%) were children under 5 years in Asembo.

Overall, verbal autopsy (VA) interviews were successfully conducted on 206 (72.3%) and 958 (93.4%) of pre-COVID-19 period registered deaths in Kibera and Asembo, respectively (Table 2). During the COVID-19 period, verbal autopsy (VA) interviews were successfully conducted on 93 (78.8%) and 366 (82.2%) of registered deaths in Kibera and Asembo, respectively.

### Circulation of SARS-CoV-2 in Asembo and Kibera PBIDS

Surveillance for COVID-19 in both sites started on 1st May 2020. The first COVID-19 cases were detected in May in Kibera (Fig 1) and September in Asembo (Fig 2). In Kibera, five spikes in SARS-CoV-2 positivity were reported in May and November of 2020, and in March, July and December of 2021. In Asembo, three spikes in SARS-CoV-2 positivity were reported in November 2020, May 2021 and December 2021.

**Table 1. Annual number of deaths and mortality rates by age and sex in Kibera and Asembo population-based infectious disease surveillance sites in Kenya, January 2016 to December 2021.**

| Characteristic | Categories | 2016 | | | 2017 | | | 2018 | | | 2019 | | | 2020 | | | 2021 | | |
|---|---|---|---|---|---|---|---|---|---|---|---|---|---|---|---|---|---|---|---|
| | | Deaths | pyo | Rate[1] | Deaths | pyo | Rate | Deaths | pyo | Rate | Deaths | pyo | Rate | Deaths | pyo | Rate | Deaths | pyo | Rate |
| **A. Kibera** | | | | | | | | | | | | | | | | | | | |
| Age groups in years, y | <1y | 10 | 487.0 | 20.5 | 12 | 588.2 | 20.4 | 13.0 | 616.6 | 21.1 | 20 | 620.0 | 32.3 | 10 | 504.9 | 19.8 | 16 | 493.4 | 32.4 |
| | 1–4 y | 5 | 1995.8 | 2.5 | 8 | 2379.1 | 3.4 | 4.0 | 2403.9 | 1.7 | 4 | 2469.1 | 1.6 | 3 | 2432.2 | 1.2 | 1 | 2167.7 | 0.5 |
| | 5–14 y | 2 | 5632.0 | 0.4 | 8 | 6497.0 | 1.2 | 4 | 6507.2 | 0.6 | 7 | 6548.8 | 1.1 | 2 | 6344.9 | 0.3 | 2 | 5853.0 | 0.3 |
| | 15–49 y | 26 | 10369.3 | 2.5 | 45 | 12433.4 | 3.6 | 31 | 12886.5 | 2.4 | 28 | 13400.2 | 2.1 | 28 | 13474.3 | 2.1 | 31 | 13036.7 | 2.4 |
| | 50–64 y | 7 | 829.6 | 8.4 | 9 | 1009.2 | 8.9 | 11 | 1095.2 | 10.0 | 8 | 1177.4 | 6.8 | 14 | 1243.3 | 11.3 | 11 | 1226.8 | 9.0 |
| | ≥65 y | 4 | 83.1 | 48.1 | 2 | 111.6 | 17.9 | 3 | 119.2 | 25.2 | 3 | 141.6 | 21.2 | 4 | 156.5 | 25.6 | 5 | 164.3 | 30.4 |
| Sex at birth | Female | 23 | 10028.0 | 2.3 | 40 | 11835.8 | 3.4 | 24 | 12161.1 | 2.0 | 34 | 12535.4 | 2.7 | 21 | 12365.7 | 1.7 | 31 | 11806.2 | 2.6 |
| | Male | 31 | 9369.9 | 3.3 | 44 | 11184.1 | 3.9 | 42 | 11467.6 | 3.7 | 36 | 11822.3 | 3.0 | 40 | 11790.3 | 3.4 | 35 | 11135.6 | 3.1 |
| Overall | All | 54 | 19397.8 | 2.8 | 84 | 23020.0 | 3.6 | 66 | 23628.7 | 2.8 | 70 | 24357.8 | 2.9 | 61 | 24156.0 | 2.5 | 66 | 22941.8 | 2.9 |
| **B. Asembo** | | | | | | | | | | | | | | | | | | | |
| Age groups in years, y | <1y | 30 | 796.4 | 37.7 | 31 | 752.4 | 41.2 | 30 | 729.5 | 41.1 | 30 | 808.3 | 37.1 | 26 | 761.2 | 34.2 | 24 | 856.7 | 28.0 |
| | 1–4 y | 26 | 3220.7 | 8.1 | 10 | 3205.4 | 3.1 | 9 | 3155.6 | 2.9 | 18 | 3116.6 | 5.8 | 18 | 2689.9 | 6.7 | 12 | 3250.4 | 3.7 |
| | 5–14 y | 10 | 9302.4 | 1.1 | 5 | 9409.1 | 0.5 | 9 | 9435.4 | 1.0 | 6 | 9442.2 | 0.6 | 11 | 7873.4 | 1.4 | 11 | 9182.2 | 1.2 |
| | 15–49 y | 70 | 13717.4 | 5.1 | 65 | 14075.8 | 4.6 | 55 | 14456.3 | 3.8 | 39 | 14895.3 | 2.6 | 50 | 10864.6 | 4.6 | 57 | 13801.7 | 4.1 |
| | 50–64 y | 30 | 2556.9 | 11.7 | 30 | 2551.1 | 11.8 | 34 | 2539.4 | 13.4 | 27 | 2596.9 | 10.4 | 37 | 1111.2 | 33.3 | 35 | 1267.1 | 27.6 |
| | ≥65 y | 101 | 1718.8 | 58.8 | 96 | 1753.9 | 54.7 | 102 | 1859.7 | 54.8 | 121 | 1944.2 | 62.2 | 105 | 261.4 | 401.7 | 100 | 330.5 | 302.5 |
| Sex at birth | Female | 132 | 16464.9 | 8.0 | 121 | 16668.7 | 7.3 | 112 | 16908.5 | 6.6 | 132 | 17216.8 | 7.7 | 129 | 11868.2 | 10.9 | 119 | 14430.1 | 8.2 |
| | Male | 135 | 14847.8 | 9.1 | 116 | 15079.0 | 7.7 | 127 | 15267.2 | 8.3 | 109 | 15586.6 | 7.0 | 118 | 11693.5 | 10.1 | 120 | 14258.6 | 8.4 |
| Overall | All | 267 | 31312.7 | 8.5 | 237 | 31747.7 | 7.5 | 239 | 32175.7 | 7.4 | 241 | 32803.4 | 7.3 | 247 | 23561.7 | 10.5 | 239 | 28688.7 | 8.3 |

Key:1, mortality rate per 1,000 person-years of observation.

**Table 2. Number of deaths, proportion of deaths with verbal autopsy, mortality rates and incidence rate ratios stratified by age and sex for pre- and during COVID-19 period in Kibera and Asembo, Kenya.**

| Characteristics | Categories | Total deaths | | % of deaths with VA | | All-cause mortality | | | | | | Incidence Rate Ratio[1] (95% CI) | P-value |
|---|---|---|---|---|---|---|---|---|---|---|---|---|---|
| | | Pre-COVID | During-COVID | Pre-COVID | During-COVID | Pre-COVID | | | During-COVID | | | | |
| | | n | n | % | % | n | pyo | rate | n | pyo | rate | | |
| **A. Kibera** | | | | | | | | | | | | | |
| Age groups in years, y | <1y | 56 | 25 | 57.1 | 88.0 | 56 | 2399.5 | 23.3 | 25 | 910.595 | 27.5 | 1.18 (0.73–1.88) | 0.499 |
| | 1–4 y | 22 | 3 | 77.3 | 100.0 | 22 | 9648.5 | 2.3 | 3 | 4199.4 | 0.7 | 0.31 (0.09–1.05) | 0.059 |
| | 5–14 y | 21 | 4 | 85.7 | 100.0 | 21 | 26232.5 | 0.8 | 4 | 11150.4 | 0.4 | 0.32 (0.10–1.07) | 0.065 |
| | 15–49 y | 135 | 54 | 74.8 | 79.6 | 135 | 51266.0 | 2.6 | 54 | 24334.4 | 2.2 | 0.84 (0.61–1.16) | 0.288 |
| | 50–64 y | 38 | 23 | 71.1 | 60.9 | 38 | 4308.6 | 8.8 | 23 | 2272.9 | 10.1 | 1.10 (0.65–1.89) | 0.728 |
| | ≥65 y | 13 | 9 | 84.6 | 77.8 | 13 | 480.7 | 27.0 | 9 | 295.6 | 30.4 | 1.00 (0.41–2.41) | 0.999 |
| Sex at birth | Female | 125 | 48 | 72.8 | 79.2 | 125 | 48583.1 | 2.6 | 48 | 22149.1 | 2.2 | 0.84 (0.60–1.17) | 0.312 |
| | Male | 160 | 70 | 71.9 | 78.6 | 160 | 43843.9 | 3.6 | 70 | 22926.0 | 3.1 | 0.94 (0.71–1.24) | 0.647 |
| Overall | All | 285 | 118 | 72.3 | 78.8 | 285 | 92427.0 | 3.1 | 118 | 45075.1 | 2.6 | 0.88 (0.71–1.09) | 0.242 |
| **B. Asembo** | | | | | | | | | | | | | |
| Age groups in years, y | <1y | 125 | 46 | 89.6 | 58.7 | 125 | 3190.7 | 39.2 | 46 | 1513.82 | 30.4 | 0.78 (0.55–1.09) | 0.141 |
| | 1–4 y | 65 | 26 | 87.7 | 80.8 | 65 | 13091.7 | 5.0 | 26 | 5546.9 | 4.7 | 0.92 (0.58–1.44) | 0.704 |
| | 5–14 y | 34 | 20 | 88.2 | 85.0 | 34 | 38771.6 | 0.9 | 20 | 15873.0 | 1.3 | 1.53 (0.87–2.67) | 0.138 |
| | 15–49 y | 233 | 100 | 89.3 | 77.0 | 233 | 58684.5 | 4.0 | 100 | 23126.6 | 4.3 | 1.09 (0.86–1.38) | 0.466 |
| | 50–64 y | 121 | 68 | 97.5 | 91.2 | 121 | 10413.1 | 11.6 | 68 | 2209.6 | 30.8 | 2.62 (1.95–2.52) | **<0.001** |
| | ≥65 y | 447 | 185 | 96.9 | 87.6 | 447 | 7314.8 | 61.1 | 185 | 553.7 | 334.1 | 5.47 (4.60–6.50) | **<0.001** |
| Sex at birth | Female | 516 | 229 | 93.6 | 81.2 | 516 | 68989.6 | 7.5 | 229 | 24567.7 | 9.3 | 1.25 (1.07–1.46) | **0.006** |
| | Male | 509 | 216 | 93.3 | 83.3 | 509 | 62476.7 | 8.1 | 216 | 24255.9 | 8.9 | 1.09 (0.93–1.28) | 0.273 |
| Overall | All | 1025 | 445.0 | 93.5 | 82.2 | 1025 | 131466.3 | 7.8 | 445 | 48823.6 | 9.1 | 1.16 (1.04–1.31) | **0.006** |

Key: 1, Incidence rate ratios comparing rates during with those in the pre-COVID-19 period

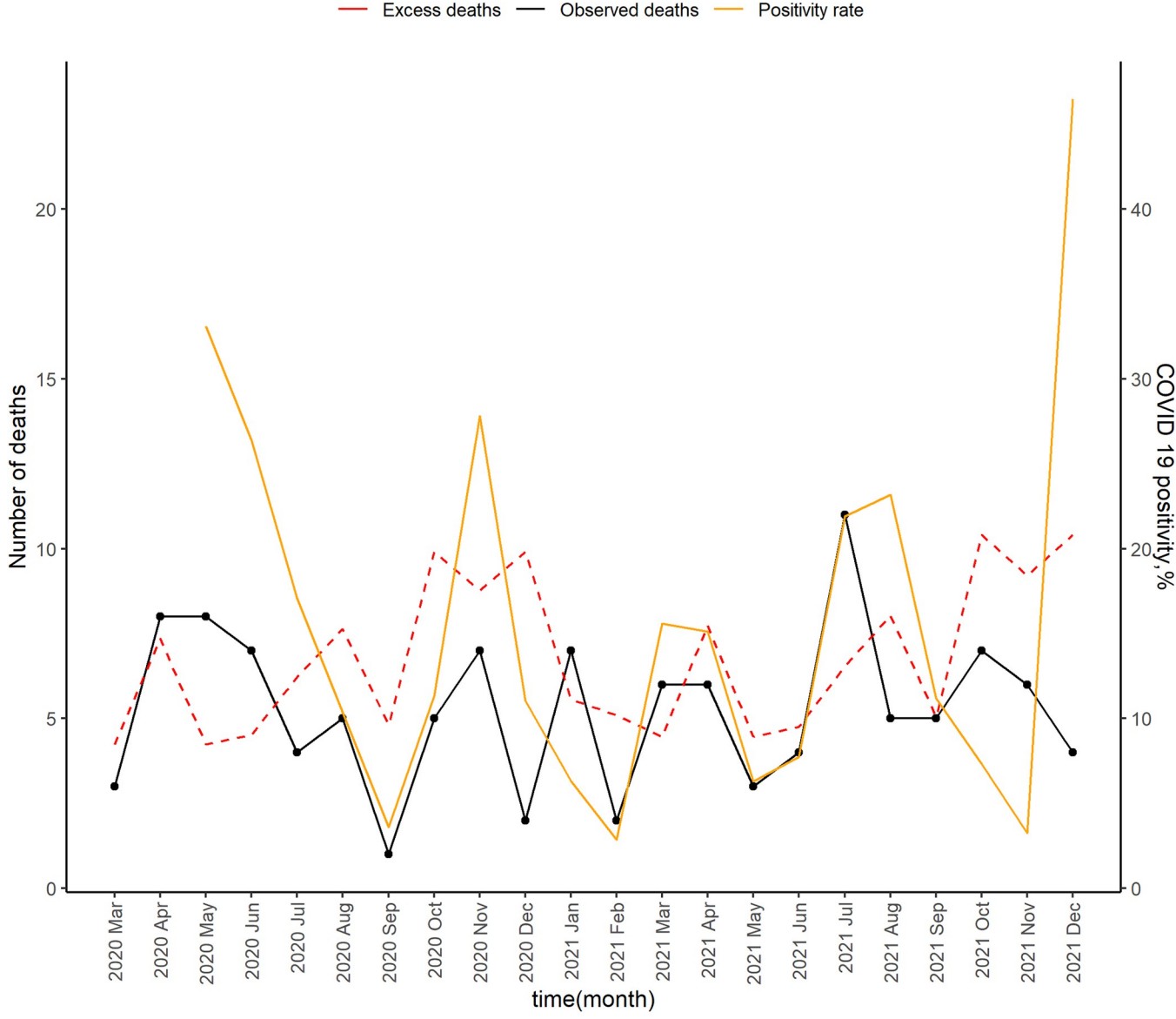

**Fig 1. Number of deaths observed, expected deaths and the SARS-CoV 2 PCR positivity rate as reported in Kibera, Kenya, March 2020-Dec 2021.**

## All-cause mortality rates before and during COVID-19 in Asembo and Kibera PBIDS

In Kibera, the all-cause mortality rate was 3.1 (95% Confidence Interval [CI], 2.7–3.4) per 1000 person–years (pyo) while in Asembo the all-cause mortality rate was 7.8 [95% CI 7.3–8.3] per 1000 pyo in the pre-COVID-19 period. The corresponding all-cause mortality rates during the COVID-19 period were 2.6 [95% CI, 2.2–3.2] and 9.1 [95% CI, 8.2–10.0] per 1000 pyo in Kibera and Asembo, respectively (Table 2). Though the all-cause mortality was lower in the COVID-19 period in Kibera relative to pre-COVID-19 period, the difference was not statistically significant (incidence rate ratio, IRR, 0.88; 95% CI, 0.71–1.09; p-value = 0.242). However, a statistically significant increase in the all-cause mortality rate was observed (IRR, 1.16; 95% CI, 1.04–1.31; p = 0.006) in Asembo. In Asembo, a statistically significant increase in all-cause

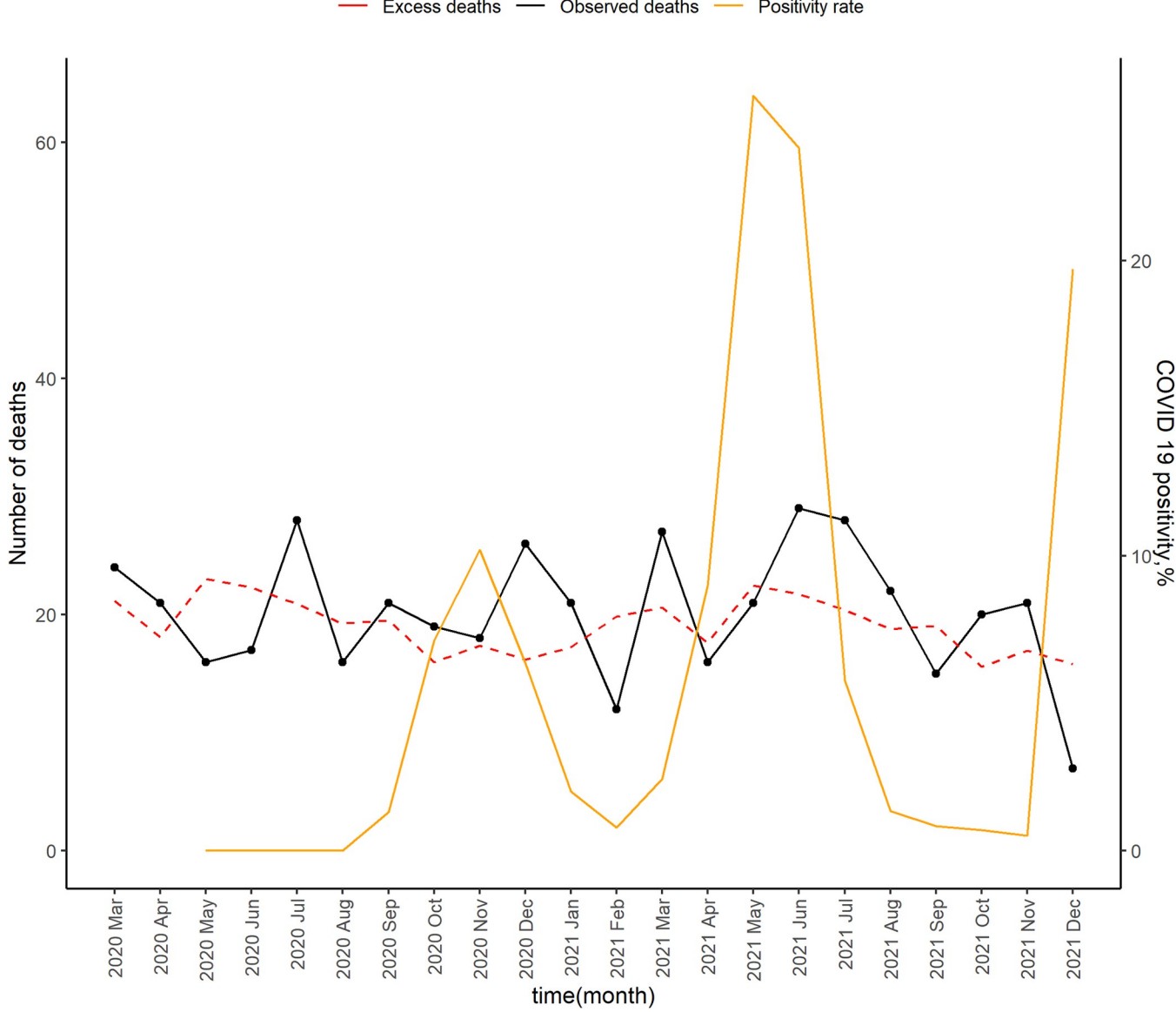

**Fig 2. Number of deaths observed, expected deaths and the SARS-CoV 2 PCR positivity rate as reported in Asembo, Kenya, March 2020- Dec 2021.**

mortality rate was observed among persons aged 50 to 64 years (IRR, 2.62; 95% CI, 1.95–3.52; p<0.001), persons aged 65 years and above (IRR, 5.47; 95% CI, 4.60–6.50; p<0.001) and among females (IRR, 1.25; 95% CI, 1.07–1.46; p = 0.006) in the COVID-19 period compared to the pre-COVID-19 period. In Kibera, the all-cause mortality rate was higher among infants (<1 years) during the COVID-19 period compared to in the pre-COVID-19 period though not statistically significant (IRR, 1.18; 95% CI, 0.73–1.88; p = 0.499). All-cause mortality rate among persons aged 50 to 64 years was 10.1 per 1000 pyo during the COVID-19 period compared to 8.8 per 1000 pyo in the pre-COVID-19 period (IRR, 1.10; 95% CI, 0.65–1.89; p = 0.728). Table 2 shows a summary of the number of deaths, proportion of deaths with verbal autopsy and all-cause mortality rates by age and sex in both sites in the pre- and during the COVID-19 period.

### P-score estimates during COVID-19 in Asembo and Kibera PBIDS

Overall, during the COVID-19 period the observed deaths were higher than expected deaths in Asembo (P-score = 6.0%) and lower than expected in Kibera (P-score = -22.3%) (Table 3). Males had a P-score of 9.6% and -5.4% in Asembo and Kibera, respectively. Females had a P-score of 8.4% and -27.3% in Asembo and Kibera, respectively. Children under 5 years had a P-score of 13.5% and -47.6% in Asembo and Kibera, respectively. Older children and adults ($\geq$5 years) had a P-score of 8.6% and -12.6% in Asembo and Kibera, respectively.

In Kibera, there was a moderate positive correlation between positivity rate and reported p-score, $r_s$ = 0.33; however, the relationship was not significant (p = 0.160). In Asembo, there was a weak negative correlation between positivity rate and reported p-score, $r_s$ = -0.14; however, the relationship was not significant (p = 0.548)

### Causes of death in Asembo and Kibera PBIDS

In Kibera, the highest observed change in proportion of deaths were due to acute respiratory infection including pneumonia and road traffic accidents. The proportion of deaths due to acute respiratory infection including pneumonia during the COVID-19 period was 5.4% compared to 13.1% in the pre-COVID-19 period (Fig 3).

The proportion of deaths due to road traffic accidents was 9.7% during the COVID-19 period and 6.3% in the pre-COVID-19 period. In Asembo, the highest observed change in proportion of deaths were due to acute respiratory infection including pneumonia and pulmonary tuberculosis. The proportion of deaths due to acute respiratory infection including pneumonia during COVID-19 period was 19·1% compared to 14·3% in the pre-COVID-19 period (Fig 4).

The proportion of deaths due to pulmonary tuberculosis during COVID-19 period was 3.8% compared to 7·9% in the pre-COVID-19 period. Overall, there was no evidence of change in mortality rate due to top 3 age group specific causes of death in the pre-COVID-19 period in Kibera. However, in Asembo, among persons aged 45 years and above there was an increase in mortality rate due to acute respiratory infection including pneumonia (incidence rate ratio (IRR),1.45;95% CI, 1.03–2.04; p = 0.031). A reduction in mortality rate due to pulmonary tuberculosis (0.22;95% CI, 0.05–0.87; p = 0.031) was observed in the 5–44year age group (Table 4).

## Discussion

Overall, our findings suggest higher-than-expected deaths in the rural surveillance site during the first 22 months of the pandemic in Kenya. Globally, an excess mortality of 14.9 million was reported in 2020 and 2021 with lower- and middle-income countries accounting for 53% of the excess deaths [24]. A study tracking excess mortality across 103 countries during the COVID-pandemic found that in several worst-affected countries (Peru, Ecuador, Bolivia, Mexico) the excess mortality was above 50% of the expected annual mortality. The study reported an excess mortality of 32% in South Africa the only country in Africa included in the study [25].

Our study showed an increase in all-cause mortality rates during the COVID-19 period in the rural site in western Kenya. This increase was observed among women and persons aged 45 years and above. However, there was no evidence of change in all-cause mortality rates in the urban informal settlement surveillance site in Kibera, in the capital city of Kenya—Nairobi. The observed increase in all-cause mortality rate corroborates findings from a study conducted in the year 2020 in rural Bangladesh which reported higher mortality rate among persons aged 65 years and above during the COVID-19 period compared to previous years. However, they observed no difference in any of the younger age-groups [26]. In rural areas, several

**Table 3. Observed deaths, expected deaths, excess deaths, P-scores and COVID-19 positivity during the COVID-19 period in Kibera and Asembo, Kenya, March 2020 to December 2021.**

| | Kibera | | | | | Asembo | | | | |
|---|---|---|---|---|---|---|---|---|---|---|
| Month | Expected number of deaths (95% CI) | Observed number of deaths | Excess deaths | P-score, % | COVID-19 Positivity, % | Expected number of deaths (95% CI) | Observed number of deaths | Excess deaths | P-score, % | COVID-19 Positivity, % |
| 2020 Mar | 4 (1–10 | 3 | -1 | -29.3 | | 21 (13–32) | 24 | 3 | 13.5 | |
| 2020 Apr | 7 (3–14) | 8 | 1 | 8.8 | | 18 (11–28) | 21 | 3 | 16.1 | |
| 2020 May | 4 (1–10) | 8 | 4 | **88.6** | **32.0** | 23 (15–35) | 16 | -7 | -30.5 | 0.0 |
| 2020 Jun | 5 (2–12) | 7 | 2 | 54.7 | 26.4 | 22 (14–33) | 17 | -5 | -23.8 | 0.0 |
| 2020 Jul | 6 (2–13) | 4 | -2 | -35.7 | 17.2 | 21 (13–32) | 28 | 7 | 33.9 | 0.0 |
| 2020 Aug | 8 (3–16) | 5 | -3 | -34.5 | 10.4 | 19 (11–30) | 16 | -3 | -16.9 | 0.0 |
| 2020 Sep | 5 (2–12) | 1 | -4 | -79.2 | 3.6 | 19 (11–30) | 21 | 2 | 7.7 | 1.3 |
| 2020 Oct | 10 (5–18) | 5 | -5 | -49.5 | 11.3 | 16 (9–26) | 19 | 3 | 19.0 | 7.1 |
| 2020 Nov | 9 (4–17) | 7 | -2 | -20.2 | **27.7** | 17 (10–27) | 18 | 1 | 3.6 | **10.2** |
| 2020 Dec | 10 (5–18) | 2 | -8 | **-79.8** | 10.2 | 16 (9–26) | 26 | 10 | **60.4** | 6.4 |
| 2021 Jan | 6 (2–13) | 7 | 1 | 25.9 | 6.3 | 17 (10–27) | 21 | 4 | 21.7 | 2.0 |
| 2021 Feb | 5 (2–12) | 2 | -3 | -60.8 | 2.9 | 20 (12–31) | 12 | -8 | -39.6 | 0.8 |
| 2021 Mar | 4 (1–10) | 6 | 2 | 34.5 | 15.6 | 21 (13–32) | 27 | 6 | 30.9 | 2.4 |
| 2021 Apr | 8 (3–16) | 6 | -2 | -22.4 | 14.6 | 18 (11–28) | 16 | -2 | -9.3 | 9.0 |
| 2021 May | 4 (1–10) | 3 | -1 | -32.7 | 6.7 | 22 (14–33) | 21 | -1 | -6.5 | 25.6 |
| 2021 Jun | 5 (2–12) | 4 | -1 | -15.9 | 7.5 | 22 (14–33) | 29 | 7 | 33.2 | 23.8 |
| 2021 Jul | 7 (3–14) | 11 | 4 | 68.2 | 22.4 | 20 (12–31) | 28 | 8 | 37.3 | 5.8 |
| 2021 Aug | 8 (3–16) | 5 | -3 | -37.7 | 23.1 | 19 (11–30) | 22 | 3 | 17.1 | 1.4 |
| 2021 Sep | 5 (2–12) | 5 | 0 | -1.1 | 11.3 | 19 (11–30) | 15 | -4 | -21.1 | 0.8 |
| 2021 Oct | 10 (5–18) | 7 | -3 | -32.7 | 7.2 | 16 (9–26) | 20 | 4 | 28.4 | 0.7 |
| 2021 Nov | 9 (4–17) | 6 | -3 | -34.9 | 3.0 | 17 (10–27) | 21 | 4 | 23.9 | **0.5** |
| 2021 Dec | 10 (5–18) | 4 | -6 | -61.6 | 46.5 | 16 (9–26) | 7 | -9 | **-55.7** | 19.7 |
| Total | 149 (117–164) | 116 | **-33** | **-22.3** | 13.6 | 420 (370–451) | 445 | **25** | **6.0** | 6.9 |

determinants including high poverty rates, low health literacy levels, cultural norms, educational disparities, distance to facility have been identified as barriers to healthcare access [27, 28]. We hypothesize that the lack of change in mortality rates in our urban site could be due to

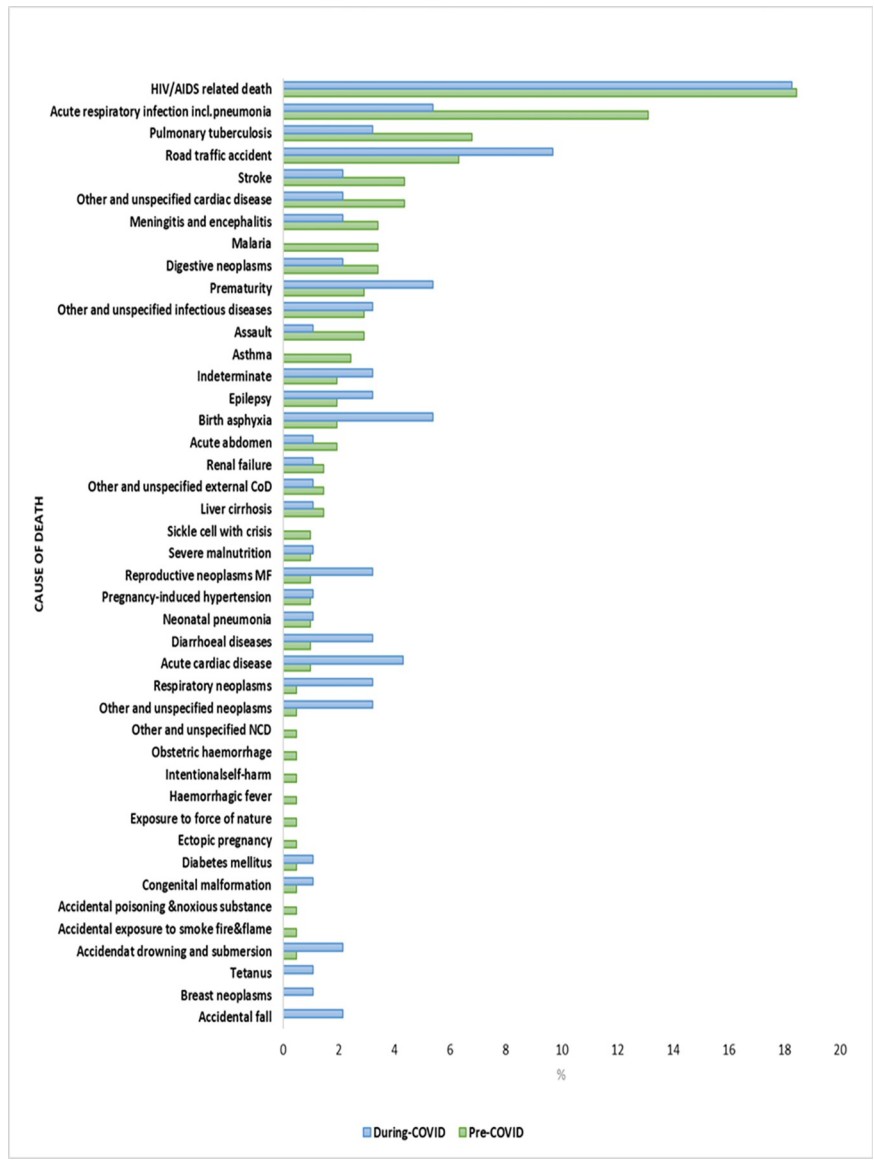

**Fig 3. Proportion of probable causes of death pre- and-during COVID-19 period in Kibera site.** NCD-Non-Communicable Diseases; CoD-Cause of Death.

multiple factors. First, the urban informal settlement had a youthful demography compared to the rural site. Factors such as young population have been hypothesized as attributing to the apparent lower than expected morbidity and mortality of COVID-19 in sub-Saharan Africa (SSA) [29]. Secondly, more enforced preventive measures such as restricted movement, lockdowns and stay-at-home orders and more adherence to non-pharmaceutical interventions like mask wearing in Nairobi might have had an effect on mortality. Non-pharmaceutical interventions such as movement restrictions have been reported to reduce transmission of SARS coronavirus 2 (SARS-CoV-2) in several countries [30]. In addition, the lack of change in mortality rates could be attributed to a relatively high COVID-19 vaccination coverage in urban areas compared to rural areas in Kenya [31]. We observed changes in mortality due to acute respiratory infection including pneumonia and pulmonary tuberculosis among older children and

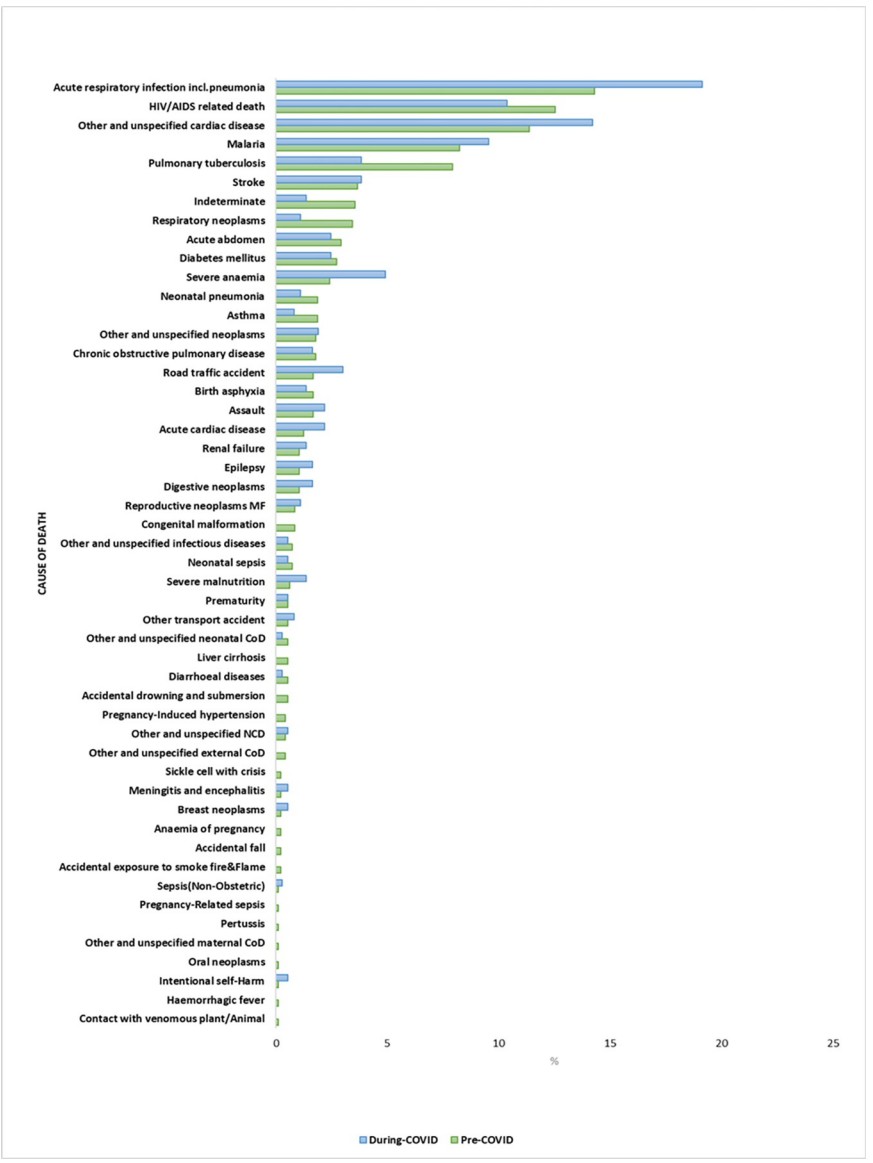

**Fig 4. Proportion of probable causes of death pre-and-during COVID-19 period in Asembo site.** NCD-Non-Communicable Diseases; CoD-Cause of Death.

adults (≥5 years) in our rural site. There are limited studies that have used verbal autopsy data to conduct a comparison of deaths attributable to specific causes pre-and-during the COVID-19 period. A study conducted in rural coastal Kenya reported a transient reduction in deaths attributable to acute respiratory infections in 2020 [32]. A few other modelling studies, though not utilizing verbal autopsy data, have predicted significant increases in deaths due to other communicable diseases such as malaria, tuberculosis and HIV/AIDS due to the effects of the COVID-19 pandemic in sub-Saharan Africa [33–35].

## Limitations of the study

Our study had a few limitations. First, there is possibility of misclassification of CoD especially with the use of VA data and the InterVA-5 model just like in all other forms of cause of death

**Table 4. Mortality rates for top 3 specific causes of death pre COVID-19 Period by age group in Kibera and Asembo.**

| Kibera | | Mortality rates | | | |
|---|---|---|---|---|---|
| Age groups in years, y | Cause of death[1] | Pre-covid | During covid | Adjusted Rate ratio (95% CI) | P-value |
| <5 y | Acute resp infect incl pneumonia | 0.16 | 0.00 | - | - |
| | Prematurity | 0.09 | 0.15 | 1.71(0.62–4.69) | 0.298 |
| | Birth asphyxia | 0.06 | 0.15 | 2.56(0.84–7.84) | 0.099 |
| 5–44 y | HIV/AIDS related death | 0.36 | 0.31 | 0.85(0.46–1.60) | 0.623 |
| | Pulmonary tuberculosis | 0.15 | 0.09 | 0.62(0.21–1.84) | 0.385 |
| | Acute resp infect incl pneumonia | 0.12 | 0.12 | 1.03(0.37–2.84) | 0.962 |
| ≥45 y | HIV/AIDS related death | 0.15 | 0.15 | 1.03(0.41–2.55) | 0.957 |
| | Acute resp infect incl pneumonia | 0.12 | 0.03 | 0.26(0.04–1.50) | 0.131 |
| | Stroke | 0.07 | 0.03 | 0.41(0.07–2.54) | 0.339 |
| **Asembo** | | | | | |
| Age groups in years, y | | | | | |
| <5 y | Malaria | 0.32 | 0.31 | 0.97(0.54–1.74) | 0.91 |
| | Acute resp infect incl pneumonia | 0.25 | 0.20 | 0.78(0.38–1.60) | 0.501 |
| | Neonatal pneumonia | 0.15 | 0.09 | 0.60(0.21–1.71) | 0.337 |
| 5–44 y | HIV/AIDS related death | 0.42 | 0.35 | 0.83(0.48–1.42) | 0.496 |
| | Pulmonary tuberculosis | 0.20 | 0.04 | 0.22(0.05–0.87) | **0.031** |
| | Malaria | 0.20 | 0.20 | 1.01(0.48–2.12) | 0.98 |
| ≥45 y | Acute resp infect incl pneumonia | 0.72 | 1.05 | 1.45(1.03–2.04) | **0.031** |
| | HIV/AIDS related death | 0.45 | 0.37 | 0.83(0.49–1.41) | 0.494 |

ascertainment methods. Secondly, in both sites the highest change in proportion of deaths during COVID-19 period were as a result of ARI/pneumonia in both sites. However, we were unable to ascertain what proportion of these deaths were due to COVID-19 or COVID-19related. Thirdly, PBIDS participants receive health care for acute illnesses free of charge at the surveillance clinics, and this access to care might have reduced mortality rates and decreased generalizability of our results to other settings. In addition, the rural site having a more elderly population might have had an impact on the mortality rates.

## Conclusion

In conclusion, this study provides useful insights on the direct or indirect COVID-19 effect on mortality in both rural and urban informal settlements in Kenya. The findings highlight the overall burden of the pandemic and provides a baseline for the future evaluation of the public health impact of COVID-19 prevention and vaccination response efforts in diverse settings. We noted significant changes in all-cause and cause specific mortality in our sites. This is an indication of a heterogenous impact of COVID-19 on mortality across diverse populations in Kenya. Based on the p-score calculations we observed excess deaths greater than expected during the COVID-19 period in the rural site and fewer than expected deaths in the urban site.

## Supporting information

**S1 Table. Proportion of causes of death pre-and-during COVID-19 period in Kibera, Kenya, 2016–2021.** NCD-Non-Communicable Diseases; CoD-Cause of Death.
(DOCX)

**S2 Table. Proportion of causes of death pre- and during COVID-19 period in Asembo, Kenya,2016–2021.** NCD-Non-Communicable Diseases; CoD-Cause of Death.
(DOCX)

## Acknowledgments

The authors wish to thank the residents of Asembo and Kibera informal settlement for their continued participation in the surveillance. We would wish also to acknowledge the following teams for their contributions to this study; The field workers for their dedication in collecting this data, the data team for data management and field supervisors for supervision of field activities. This paper is published with permission from the Director General of the Kenya Medical Research Institute.

## Author Contributions

**Conceptualization:** Clifford Oduor, Godfrey Bigogo, Patrick K. Munywoki.

**Data curation:** Clifford Oduor, Allan Audi, Samwel Kiplangat, Joshua Auko.

**Formal analysis:** Clifford Oduor.

**Methodology:** Clifford Oduor.

**Supervision:** Patrick K. Munywoki.

**Visualization:** Clifford Oduor.

**Writing – original draft:** Clifford Oduor.

**Writing – review & editing:** Alice Ouma, George Aol, Carolyne Nasimiyu, George O. Agogo, Terrence Lo, Peninah Munyua, Amy Herman-Roloff, Godfrey Bigogo, Patrick K. Munywoki.

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
