## [Decision Letter · Decision Letter 0]

6 Jun 2023

PGPH-D-23-00661

Estimating excess mortality during the COVID-19 pandemic from a population-based infectious disease surveillance in two diverse populations in Kenya, March 2020-December 2021

Dear Dr. Oduor,

Thank you for submitting your manuscript to PLOS Global Public Health. After careful consideration, we feel that it has merit but does not fully meet PLOS Global Public Health’s publication criteria as it currently stands. Therefore, we invite you to submit a revised version of the manuscript that addresses the points raised during the review process.

We look forward to receiving your revised manuscript.

Kind regards,

Humayun Kabir

Academic Editor

Journal Requirements:

Additional Editor Comments (if provided):

Reviewers' comments:

Reviewer's Responses to Questions

**Comments to the Author**

1. Does this manuscript meet PLOS Global Public Health’s publication criteria? Is the manuscript technically sound, and do the data support the conclusions? The manuscript must describe methodologically and ethically rigorous research with conclusions that are appropriately drawn based on the data presented.

Reviewer #1: Yes

Reviewer #2: Yes

Reviewer #3: Yes

Reviewer #4: Yes

Reviewer #5: Yes

2. Has the statistical analysis been performed appropriately and rigorously?

Reviewer #1: Yes

Reviewer #2: Yes

Reviewer #3: I don't know

Reviewer #4: Yes

Reviewer #5: Yes

3. Have the authors made all data underlying the findings in their manuscript fully available (please refer to the Data Availability Statement at the start of the manuscript PDF file)?

Reviewer #1: Yes

Reviewer #2: Yes

Reviewer #3: Yes

Reviewer #4: Yes

Reviewer #5: Yes

4. Is the manuscript presented in an intelligible fashion and written in standard English?

Reviewer #1: Yes

Reviewer #2: Yes

Reviewer #3: Yes

Reviewer #4: Yes

Reviewer #5: Yes

5. Review Comments to the Author

Reviewer #1: Excellent manuscript. I applaud the research and the work that has been put into conducting this study. Excellently worded manuscript. Please see my comments below.

Results Section:

1. Do we know what percent of the all cause mortality rates are attributed to COVID-19? For example - for people who died of respiratory illnesses, do we know what percent of these deaths are a side effect of COVID-19 related illnesses?

2. How does the increase in vaccination rates reflect on the mortality rates? Do we anticipate a reduction in the mortality rates once the vaccination rates arise?

Reviewer #2: The Manuscript is scientifically sound. easy for a leader to follow through and understand the processes in carrying out the study. The research was able to provide all the statistical methods used to analyze the data even the methods of data collection which is in line with the type of study carried out. The manuscript was written in a manner that it provides enough data to justify carrying out the research, the outcome and the limitation of the study were elaborated which is important for providing insights of the gaps that the study had. I recommend this research for publishing. However, try to explain further on the differences in mortality of the whole data as at first sight without the analysis you might conclude that the period March 2020 to 2021 had increase in deaths for the two study locations compared to the period 2016 to February 2020. if you put into considerations the time periods. The first time period had a different of 4 years to the comparison group which is just a year. I suggest explaining more of deaths in reduced time periods rather than comparing the longer period to the shorter period.

All in all, congratulations for a job well done in providing scientific data on mortality during this time period as statistics had shown that there was not much mortality due to Covid -19 in Africa Compared to other parts of the world.

Note: data for the comments above has been highlighted in the attachment.

Reviewer #3: MY dear authors many thanks for this value paper and simple writing

1- regarding the line 170 to 173 regarding specific causes of death pre COVID-19 Period by age group in Kibera and Asembo

you remember the mortality reduction at both urban and rural for pulonary tuberculosis patients during covid

i ask whats type of pulomnary is active or controlled ( closed case ) on DOT or not negative sputum or positive

so reduction at mortality regarding these pulmonary T.B during covid may be due another factors like DOT treatment , nore health care supervision or admitted patients.

2- you not remember the smoking as risk factor if has positive or negatively effect

3- the rural araea like Asembo incresed mortality may due to many factors related to education level or health facilities or tradition about covid-19 virsu or lach of mass media health information so high mortality may has many causes direct or undirect

Finally many thanks for your efforts

Reviewer #4: The 95% Confidence intervals can be written as: 95% C.I example: The all-cause mortality rate was higher during the COVID-19 period compared to the pre-COVID-19 period in Asembo [9.1, 95% CI: 8.2 – 10.0)

vs. 7.8, 95% CI: 7.3 – 8.3) per 1000 person-years of observation, pyo]

The figure showing the causes of death could be classified into Non-communicable diseases and Communicable diseases

Reviewer #5: This is a good paper. A few comments for the authors.

1. what is the design of the study?

2. What is the size of the data?

3. Did you use all the data generated in Asembo and Kibera from 2006-2021?

4. Is there any reason for choosing Verbal Autopsy interviews (using next of kin, health workers and relatives) to ascertain cause of death rather than assessing the medical records of these deceased patients?

5. Please change the tenses in the methods section to the past

6. PLOS authors have the option to publish the peer review history of their article (what does this mean?). If published, this will include your full peer review and any attached files.

**Do you want your identity to be public for this peer review?** For information about this choice, including consent withdrawal, please see our Privacy Policy.

Reviewer #1: **Yes: **Aiswarya Bulusu

Reviewer #2: **Yes: **Deborah Tembo

Reviewer #3: **Yes: **AMR Ahmed

Reviewer #4: No

Reviewer #5: No

<quillbot-extension-portal></quillbot-extension-portal>

---

## [Decision Letter · Decision Letter 1]

17 Jul 2023

Estimating excess mortality during the COVID-19 pandemic from a population-based infectious disease surveillance in two diverse populations in Kenya, March 2020-December 2021

PGPH-D-23-00661R1

Dear Dr. Oduor,

We are pleased to inform you that your manuscript 'Estimating excess mortality during the COVID-19 pandemic from a population-based infectious disease surveillance in two diverse populations in Kenya, March 2020-December 2021' has been provisionally accepted for publication in PLOS Global Public Health.

Best regards,

Humayun Kabir

Academic Editor

Reviewer Comments (if any, and for reference):

Reviewer's Responses to Questions

**Comments to the Author**

1. If the authors have adequately addressed your comments raised in a previous round of review and you feel that this manuscript is now acceptable for publication, you may indicate that here to bypass the “Comments to the Author” section, enter your conflict of interest statement in the “Confidential to Editor” section, and submit your "Accept" recommendation.

Reviewer #4: All comments have been addressed

2. Does this manuscript meet PLOS Global Public Health’s publication criteria? Is the manuscript technically sound, and do the data support the conclusions? The manuscript must describe methodologically and ethically rigorous research with conclusions that are appropriately drawn based on the data presented.

Reviewer #4: Yes

3. Has the statistical analysis been performed appropriately and rigorously?

Reviewer #4: Yes

4. Have the authors made all data underlying the findings in their manuscript fully available (please refer to the Data Availability Statement at the start of the manuscript PDF file)?

Reviewer #4: Yes

5. Is the manuscript presented in an intelligible fashion and written in standard English?

Reviewer #4: Yes

6. Review Comments to the Author

Reviewer #4: No Comments

7. PLOS authors have the option to publish the peer review history of their article (what does this mean?). If published, this will include your full peer review and any attached files.

**Do you want your identity to be public for this peer review?** For information about this choice, including consent withdrawal, please see our Privacy Policy.

Reviewer #4: **Yes: **Yao Ahonon

<quillbot-extension-portal></quillbot-extension-portal>